# Principal component analysis of coronaviruses reveals their diversity and seasonal and pandemic potential

**Tomokazu Konishi** *

Faculty of Bioresource Sciences, Akita Prefectural University, Akita, Japan

* konishi@akita-pu.ac.jp

## Abstract

Coronaviruses and influenza viruses have similarities and differences. In order to comprehensively compare them, their genome sequencing data were examined by principal component analysis. Coronaviruses had fewer variations than a subclass of influenza viruses. In addition, differences among coronaviruses that infect a variety of hosts were also small. These characteristics may have facilitated the infection of different hosts. Although many of the coronaviruses were conservative, those repeatedly found among humans showed annual changes. If SARS-CoV-2 changes its genome like the Influenza H type, it will repeatedly spread every few years. In addition, the coronavirus family has many other candidates for new pandemics.

**Data Availability Statement:** Most relevant data are available in the Supporting Information files. However, data from GISAID are available at https://www.gisaid.org/.

## Introduction

Coronavirus disease 2019 (COVID-19) is rapidly spreading worldwide [1, 2]. To investigate its spreading mechanism, genomes of coronaviruses were analysed by principal component analysis (PCA) [3] and compared to influenza viruses [4]. Both coronaviruses and influenza viruses cause annual epidemics and pandemics; however, they differ in host specificity and the rate of mutations. Comparing the diversity and changes in their genomes will help to understand the characteristics of the present pandemic. Such characteristics would be useful for confronting the virus.

Influenza and coronaviruses have RNA genomes. Both replicate the genome using their RNA-dependent RNA polymerases, which may lead to many errors [5, 6]. This characteristic has introduced variations among the viruses, including the number and size of their open reading frames (ORFs) [7]. These viruses are very different in the lengths of genome segments; while the coronavirus genome is nearly 30 kb long [6, 8, 9], the influenza genome is divided into eight segments, and the total length is nearly 14 kb [10]. Hence, compensation for the different lengths is required to facilitate comparison; to fulfil this, the values of the PCs for samples were scaled for the length; also, the values of PCs for bases are scaled for the number of samples [11].

Some classes of coronaviruses, such as human coronavirus (HCoV), cause upper respiratory tract infections in humans. The symptoms are similar to those of the common cold,

**Funding:** The author(s) received no specific funding for this work.

**Competing interests:** The authors have declared that no competing interests exist.

although they may also cause severe pneumonia [12]. They have lower infectivity than human influenza viruses. For example, a 2010–2015 study in China reported that 2.3% and 30% of patients were positive for coronavirus and influenza virus, respectively [13]; such an inferior ratio was also found in another large study [12]. Some of the strains of coronaviruses may have much higher infectivity and cause outbreaks, such as severe acute respiratory syndrome (SARS)-CoV [9, 14, 15], Middle East Respiratory Syndrome (MERS)-CoV [15–17], and SARS-CoV-2 (SCoV2) [1, 2, 16, 18, 19]. The former two cause severe symptoms, while the latter varies from asymptomatic to critical. However, although the fatality rate of COVID-19 is still being estimated, it could become five times higher than that of seasonal influenza [20] causing many complications [21]. Additionally, COVID-19 is thought to be more contagious among certain populations and age groups than influenza [21].

The corona and influenza viruses have similarities and differences in infectivity, spread ability, and symptomatology. These differences are based on their genomes, which are important for estimating how SCoV2 will act in humans.

## Materials and methods

### Data and classification

All nucleotide sequences were analysed using the same method [3]. The method gives principal components (PCs) in a form that is scaled for the length of sequences or the number of virus samples. Sequencing data were obtained from the DNA Data Bank of Japan (DDBJ) database [22]. Aligned data, obtained with DECIPHER [23] (presented in the S1–S3 Data), were further processed to observe the relationships among virus samples by using the direct PCA method [3], which can handle data under the least assumptions. The conceptual diagram of the PCA is shown in the next section (all calculations were performed in R) [24] and updated versions of the scripts can be found in GitHub (https://github.com/TomokazuKonishi/direct-PCA-for-sequences). To avoid the imbalance effect among samples, decomposition was performed by removing clusters of similar samples, for example, clusters of sequences from SARS, MERS, or SCoV2. Instead, only one sample was included from each cluster. Aligned sequences of the virus samples are available in the S1–S3 Data.

To prepare a comprehensive data set for SCoV2, 2796 full-length sequences were obtained from the Global Initiative on Sharing All Influenza Data (GISAID) database [25] and added to those used for Fig 1. Some records were preliminary and contained many uncertain bases (designated by "N"), which may be counted as indels. To cancel such artefacts, the corresponding regions were replaced with the average data, which cancels the corresponding bases from the results of the PCA. The list of subjected sequences is available in S3 Data.

### Diagram of the PCA

sequencing data matrix (nucleotide/amino acids)
    ↓ Numerical conversion
    boolean matrix
    ↓ Finding the data centre and subtraction
    centred matrix
    ↓ Singular value decomposition
    scaled PC for base and samples

To enable calculations, the sequence data is translated into numerals to create a matrix of samples and bases [3]. The matrix is then rotated by the PCA, and the centre of the rotation is the mean sequence. The direction of rotation is determined by the directions of the differences: if distinguishing variations exist and they spread the samples into two groups, the variations

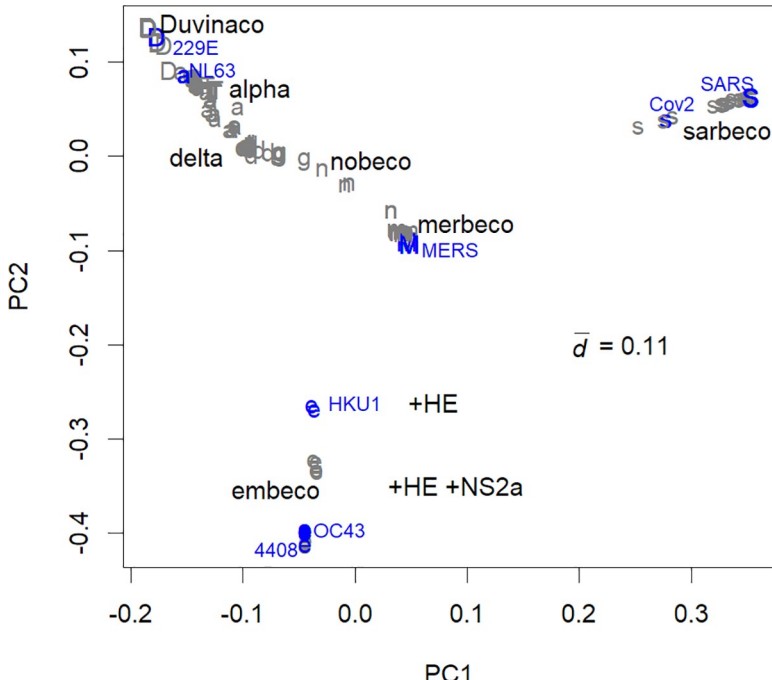

**Fig 1. Classes of coronaviruses separated into PC1 and 2.** Blue: human samples; subclasses of HCoV are indicated. a: Alphacoronavirus; d: Delta coronavirus; D: Duvinacovirus; e: Embecovirus; g: Gammacoronavirus; m: Merbecovirus; n: Nobecovirus; s: Sarbecovirus; T: TGEV. A major estimated class, Betacoronavirus, is not shown here, since the class has to cover distinctive classes (Embecovirus, Merbecovirus, Norbecovirus, and Sarbecovirus). It should be noted that many of the classifications in the original records were different from those obtained in this study.

will define a direction. If another set of variations separates the samples into a third and fourth groups, then the variations define another direction. Then, an axis is set toward a direction. The rotated data are dispersed among the axes. The axis has a sensitivity to specific nucleotide base positions, and each sample shows a specific PC for the axis. Hence, the virus samples are separated on the axis in accordance with the type of base the sample has. As all the axes are orthogonal, a smaller number of axes extract the differences efficiently. Singular value decomposition determines the direction of the axis. As the translation and rotation are reversible, the original sequences can be restored from the PCs. No unverifiable assumptions were used. These characteristics—preservation of information and falsifiability—are quite different from hierarchical clustering methods. Here, I presented the first two axes, which covered 10% to 30% of the differences in the data. The PCs were scaled to enable comparisons among different nucleotide lengths or sample numbers [11].

## Estimation of the magnitude of sample variations

The variation magnitude among sample sequences was estimated by the mean distances, scaled by the length of the sequence $m$, of virus types. This was a sort of standard deviation, $\bar{d} = \sqrt{\sum (x_i - \bar{x})^2 / 2nm}$, where $x_i$ is the Boolean of each sample sequence, $\bar{x}$ is the mean sample, and $n$ is the number of samples. Coefficient 2 was used to correct the double counts of the differences. The unit of length is the same as that of the PCA, which will extract the length toward particular directions.

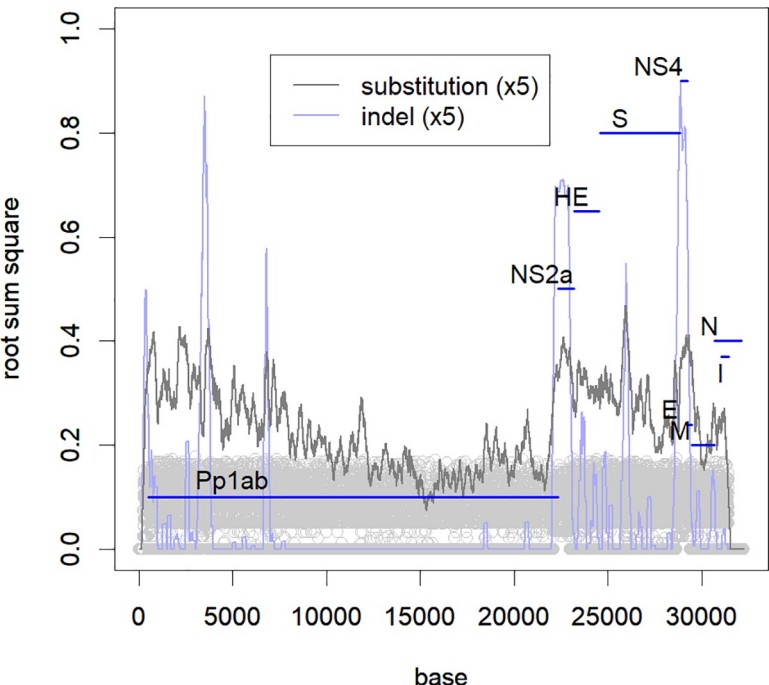

**Fig 2. Frequency of mutations in each position of the genome.** The level is estimated by the root sum square of PC1-PC5 (grey circles). The moving averages are represented in a 1:5 scale. The names of the ORFs are indicated. The indel at NS2a reflects that there are subclasses with the ORF and without the ORF.

### Estimation of mutation levels in the genome

The levels of PCs 1–5 for bases were estimated by the root sum square at each base position (Fig 2 and S3 Fig). If there are alterations in several samples, and if they occur coincidently, they may contribute to a higher level of PC. To see the tendencies at the positions, two moving averages with a width of 200 amino acid residues were shown for substitutions (grey) and indels (blue).

## Results

The coronaviruses separated into distinct classes (Fig 1), which could be further divided into subclasses (S1 Fig). For example, SARS-CoV and SCoV2 belong to different subclasses of Sarbecovirus (S1 Fig and S1 Table). The origin of the graph (0, 0) coincides with the mean data. The accumulation of mutations form a variety of viruses that have different directions and distances from the original (ancestral) virus. If the mutations and samplings are random, the original virus will be near the data mean.

The variation magnitude, estimated by the mean distance $\bar{d}$, was 0.11. This is much smaller than those of single subclasses of influenza A virus (S2 Fig), such as H1 or H9. The value has been scaled so it has a kind of generality; actually, the value of $\bar{d}$ was not significantly altered by artificial reductions of sample numbers or sequence length (not shown).

Among the classes of the coronaviruses, Gammacoronavirus and Deltacoronavirus, which are close to the origin of the graph, were mainly found in bird samples (Fig 1 and S1 Table) [6]. These could be the origin of coronaviruses, as like the influenza viruses are thought to

have originated from those of waterfowls [10]. The bat coronavirus Norbecovirus, which seems to be the origin of the mammalian viruses, represented the mean of the studied samples. Indeed, many other classes were found in bat samples (Fig 1 and S1 Table). The most distant classes from the mean, TGEV (Transmissible gastroenteritis virus) and Embecovirus, were observed to be present in larger animals and murine [6] but not in bats.

Strains of HCoV belonged to Embecovirus, Alphacoronavirus, and Dubinacovirus (Fig 1, blue). On the other hand, the strains of recent major epidemics originated from two classes without common cold viruses: Merbecovirus and Sarbecovirus.

Similar to other RNA viruses [5], many indels were observed, especially in some smaller ORFs (Fig 2 and S3 Fig). The indels ranged from small regions without frameshifts to large ones that alter multiple ORFs, for example Embecovirus is unique because it possesses an ORF of hemagglutinin. The class is further distinguished by having another ORF, NS2a, which can also not be present (Figs 1 and 2) [6]. Even within a small group of HCoVs, OC43, there was an indel corresponding to 14 amino acids in the spike protein. The classification was not significantly affected by focusing on the indels or on the rest of the sequences (S4 Fig). Therefore, indels were not given extra weight in this study; they were treated as a base or a residue. Note that some small ORFs, such as the envelope and nucleocapsid, are conservative and lack indels.

The values of PCs were not significantly affected by the hosts (Fig 1). For example, differences between bird and swine coronaviruses in Deltacoronaviruses were small (S1 Table, PC18). This is in contrast to influenza viruses, which were separated among different hosts. For example, in influenza H1N1, the waterfowl class is near the centre, with three swine groups around it, and two human groups further apart (S2 Fig) [4]. For coronaviruses, those that are more distant from Norbecovirus seem to infect larger animals, but this rule is not absolute (S1 Table).

Each of the human epidemic coronaviruses had similar viruses in bats or camels, although there were minor differences (Fig 1 and S1 Table). In the SARS-CoV spike protein, no amino acid residue was unique to humans. This is partially because our knowledge about the viruses has increased after the efforts to screen for likely viruses in wild animals [9, 26–28]. Only 35 out of 2412 residues were different between SARS-CoV and similar bat viruses, and many of these were not conserved among the bat samples (S2 Table). The situation was the same with SCoV2, which presented 34 unique amino acid residues (S2 Table); naturally, this uniqueness will be reduced after further research.

The annual occurrence of influenza A H1N1 and HCoVs is very different, since only one H1N1 variant spreads worldwide yearly (S2g Fig) [4], while several OC43 variants appear even within a single country (Fig 3A, S5 Fig and S3 Table). H1 variants will not return in the subsequent seasons, whereas OC43 variants appear repeatedly for a decade. However, by concentrating solely on one variant, the annual alterations became obvious (Fig 3B). As an example, a variant with a high PC1 value was selected, which was found in 1985–2000 in the USA and in 2002 in France (Fig 3A). The mutation speed was much slower than that of influenza H1N1 (S2G Fig); as the PC values are scaled, the magnitudes can be compared directly.

A comprehensive set of SCoV2 samples was separated into three directions, forming some classes (Fig 3C, S6A Fig). These could result from the spread of a mutated virus in a previously virus-free population. Various classes were found in some countries, suggesting multiple influx routes. The class closest to the data mean was that of China. Samples reported from countries far from China tend to show larger magnitudes of PCs, and vice versa (S4 Table).

Shift-type alterations are frequently found in the influenza virus [4], which could be caused by exchanging RNA segments among any two influenza viruses. This type of alteration was also observed in coronaviruses, even though the genome was not segmented. By focusing on the spike protein, coronaviruses separated (Fig 4A) in a similar way to the classification

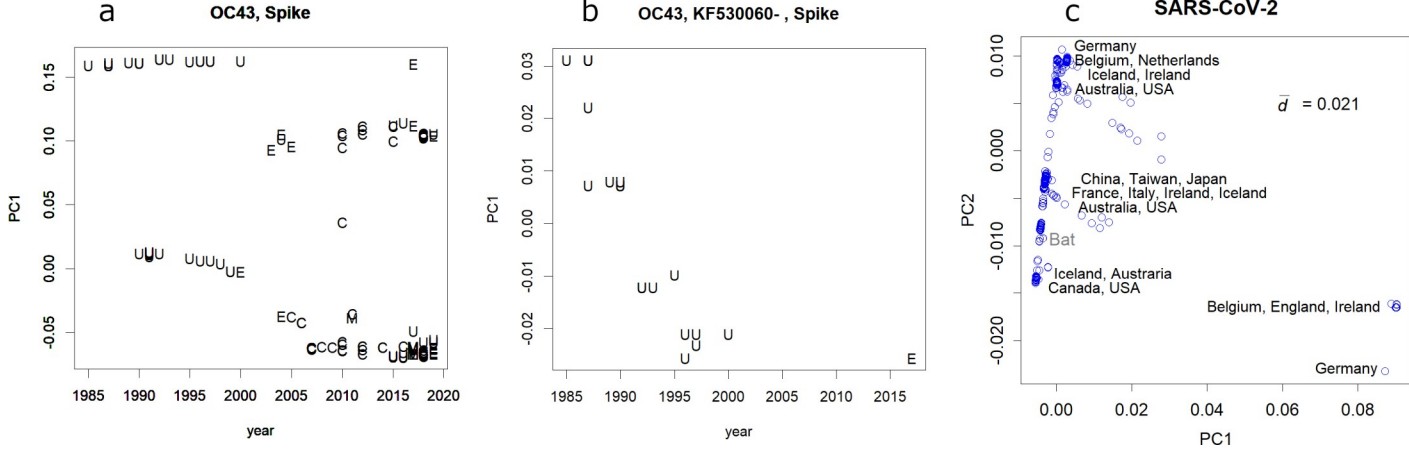

**Fig 3. a.** PC of the reported HCoV OC43 spike protein in each year. The letters represent the countries/regions: C- China, E- EU, M- Malaysia, U- USA. The upper-most series of variants were selected for a focused view (next panel). **b**, annual changes in the selected variants. Similar one-directional changes found in PC1 were always found in influenza H1N1 human cases [4]. **c**. Classes of SCoV2 separated in PC1 and 2. Data from 2836 samples are shown. Examples of countries are indicated. The full set of records is presented in S4 Table.

obtained by the whole genome (Fig 1). However, in the classification obtained by the 1ab poly-protein of coronavirus, the positions of Deltacoronavirus and Embecobirus were exchanged (Fig 4B). Additionally, the position of the nucleocapsid protein in SARS-CoV moved from OC43, losing Embecovirus unity (Fig 4C). These drastic changes can be simply explained by exchanging parts of genomes between two variants, i.e., a shift. Additionally, this phenomenon is difficult to explain without such shifts.

## Discussion

The coronavirus and influenza classes were fairly different in two aspects. First, although the groups were clearly separated, distances among the classes were much shorter in coronavirus.

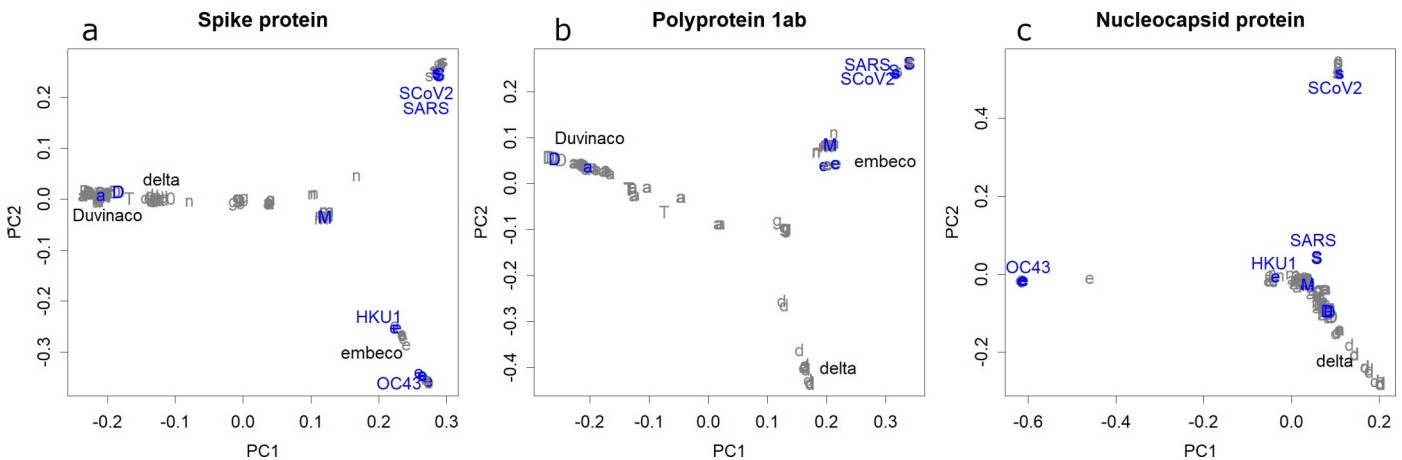

**Fig 4. a**. Classification obtained from the amino acid sequences of the spike protein. The relationships between the classes were similar to those estimated from the entire nucleotide sequences (Fig 1). **b**. Classes found in polyprotein 1ab. The positions of the Deltacoronavirus and the Sarbecovirus were replaced. **c**. Classes found in the nucleocapsid protein. SARS-CoV moved out of the Sarbecovirus class.

The variation was rather limited; the divergence magnitude among whole coronaviruses was much lower than that of a subclass of the influenza A virus. Second, distances between different hosts in coronavirus were very short; those were shorter than the short distances among the subclasses. These characteristics corroborate the assessment that "coronaviruses can apparently breach cell type, tissue, and host species barriers with relative ease" [6, 29], compared to the influenza virus, which shows clear genomic differences to different hosts [4], for example. Spike protein-mediated infection by coronavirus would be more tolerant to differences in hosts than the influenza virus haemagglutinin protein, which may have eased the selective pressure to separate the groups of coronaviruses according to the various hosts.

Shifts, which are not possible by replacing virus segments, may have occurred in coronaviruses. Since these viruses do not copy the genome into double-stranded DNA, RNA splicing in the nucleus [30] or RNA interference would be used instead of the ordinal homologous recombination by the RecA family [31]. This may also be the cause of the frequent indels.

Viruses change their genome according to various selective pressures [5], such as to: a) Maintain functions: any changes can cause malfunctions; hence, they have to conserve the genome, which is required for any state of the virus. b) Escape herd immunity: Influenza A viruses, which are highly infectious, escape herd immunity by continuously changing. All ORFs change at the same speed [4], as any part of the virus that could be represented by the major histocompatibility complex will become a target of the immune system. In contrast, HCoV is not very infectious; hence, some people remain unimmunised, easing this pressure. Additionally, the resulting smaller number of replications per year would slow the rate of changes. This might be the case for MERS-CoV among camels [15, 16, 32, 33]. c) Exploit new hosts: this may require a change in the docking system. Adaptation to the genetic system of a new host may alter codon usage and several amino acids [16]. SARS-CoV and SCoV2 might be under this type of pressure during infection. d) Increase asymptotic patients: patients with mild or no symptoms are required for survival of the virus. In humans, once all infected individuals are identified, the virus is contained, especially if the symptoms are critical.

For influenza viruses the conditions required to cause a pandemic are obvious. First, it has to be highly infective, such as type H1N1. Second, it should be free from herd immunity. For example, Pdm09 belongs to one of the two subclasses that did not cause outbreaks among humans [4]. The SARS- and MERS-CoV fulfilled these conditions, but they failed to escape the selective pressure mentioned above in (d). SCoV2 satisfies all these conditions, thus it is spreading worldwide as Pdm09 did [1, 2, 21].

For the past three decades the dominant R type H1 of influenza A changed annually [4] by changing its most variable residues, the outermost surface amino acids of the protein structure. In contrast, the inner core region of the protein was conserved; this suggests that the selective pressure is in escaping immunity. The new Pdm09 is also changing annually. Coronaviruses have shown few annual changes (Fig 3B and S6A Fig), which might be due to their limited infectibility (HCoV) or the lack of infected people (MERS-CoV). SCoV2 will face the selective pressure of herd immunity (b) as influenza A did. If it escapes this selective pressure, it will remain among humans and spread every few years. Actually, the change in SCoV2 has already begun; they have formed several classes within a short emergence time (Fig 3C, S6B Fig). The magnitudes of the PCs may indicate the migration pathways of the classes. They might mutate within China, transfer to other countries and mutate further (S6 Fig). These changes may help acclimatisation to humans (c); however, they may also relate to herd immunity (b) and/or lower lethality (d).

Fortunately, the lifespan of the classes of coronaviruses may be shorter than those of influenza viruses. The ORF lengths for influenza viruses are within a range of 2.3k-0.9Kb [10] and all ORFs change annually at the same rate [4]. In contrast, some of the coronavirus ORFs are

quite short. For example, the envelope protein [34] is only 260 bases long and located in the conserved region of the genome (Fig 2 and S3 Fig) [35]. This protein might be too short to form a variable structure, making it a good target for herd immunity. These conservative ORFs might be suitable vaccine targets. In contrast, the spike protein tended to change (S3 Fig) and may cause antibody-dependent enhancement (ADE), since it covers the surface of the virus. In ADE, the antibody that binds to the virus may help it enter the target cell through Fc or complement receptors [36] (in addition to the Ace2/Spike protein receptors [37]). There is evidence that this occurs with the SARS-CoV spike protein [38] and there is concern about SCoV2 [39]. Influenza proteins showed annual changes at the same rate, showing that any of the viral components would be under the same selective pressure [4]. If this is also true in coronaviruses, recognising proteins other than the spike protein will help with understanding the immune mechanisms.

Many bat coronaviruses seemed to be able to infect humans. The bat and human viruses are similar (Fig 1), and there are more variations in bats, of which we have observed only a part. Due to replication errors and RNA editing, a bat may possess several variants of coronavirus [5]. Although host-virus specificities are shown in laboratory experiments [29], as humans and other animals have individual variations, the barrier would be more fragile in reality. Once an infection occurs, the virus will adapt through mutations [28]. Long-term accumulation of mutations in intermediate hosts, such as pigs for influenza viruses, is not essential. These viruses would have a limited infectious character and cause mild symptoms to bats or other hosts; however, they could show an excessive adaptation to humans as SARS- and MERS-CoV. As humans do not have herd immunity to many classes of coronaviruses (other than HCoV), these might produce a new pandemic.

If intermediates are required, their main contribution in interspecies infection could be the amplification of the inoculum size and contact frequency. Bats and human habitats are different, and a bat may not produce enough viral particles to infect different hosts. The size of the donor animal is important, for example, healthcare workers with secondary MERS infections tend to have milder symptoms and a better prognosis [16]. This could be caused by differences in inoculum sizes; as camels produce nasal secretions full of viruses [32, 33], the inoculum size would be larger than that from humans. To prevent the infection of intermediate animals, live animals of different species should not be kept in the same place. Additionally, identifying the first patients is essential to prevent a human outbreak. Thus, people who have frequent contact with wildlife should not live in a cosmopolitan city.

The conventional classification system separates coronavirus into four major groups, from alpha to delta [8]; these classes have been found by a clustering method. It is true that the methodologies always find a tree structure for the relationships; however, they are not suitable for estimating the classification of many data. For example, they cannot accurately depict the relationship between distant samples (they only show the directly connected ones). Additionally, they do not suggest the root of the tree. Furthermore, they depend on many assumptions that cannot be verified. In the case of coronaviruses, the four-group structure was far from the results we found by PCA; for example, the categories of Alpha- and Beta-coronavirus actually contained large variations. Additionally, many of the credits for the classification of original sequencing records were misjudged (this was also true for influenza viruses). Estimating the classification using phylogenetic methods has poor reproducibility, and misjudgements would have been caused by such difficulty. Using an objective method is preferable to determine the attributions [3]. One of the advantages of PCA is that it can apply a classification to sets of other data without affecting the established one. This will facilitate finding attributions of new samples.

As a source of pandemic viruses, coronaviruses have many candidates that are new to humans, and can become seasonal viruses, with annual mutations. Here, the nucleotide

sequences of coronaviruses were evaluated using direct PCA [3]. The variations in coronavirus were much smaller than in influenza and differences among hosts were also small (Fig 1). There were several classes; some of them included human viruses, but others did not. SARS, MERS, and COVID-19 belonged to a class that lacked HCoVs. Some of the ORFs in coronavirus are rather conservative, and hence would be ideal targets for vaccines (Fig 2). HCoVs are more conservative than influenza (Fig 3A and S2G Fig), but they also show annual changes (Fig 3Bb). SARS-CoV-2 has changed during spreading to the world; it will continue mutating if it stays as a pandemic virus.

## Supporting information

**S1 Fig. Classes of coronaviruses at the presented axes of PCs.** Blue: human samples. Labels are the same as in Fig 1C. Classes found in the Sarbecovirus. SARS-CoV and SCoV2 belong to different groups.
(TIF)

**S2 Fig. Subclasses of influenza A virus. a**. H1 hemagglutinin, **b**. H4 PB1, **c**. H5 hemagglutinin, **d**. H7 PB2, **e**. H9 hemagglutinin, **f**. H9 PB1. Values of the mean distance were indicated. The subclass may coincide with the hosts (**d**) but in many cases, one host species formed a distinct class.
(TIF)

**S3 Fig. Levels of PC1-PC5 at each position in the nucleotide sequences. a**. Sarbecovirus, **b**. HCoV OC43, **c**. MERS-CoV. **d**. SCoV2. Names of the ORFs are indicated in Fig 2.
(TIF)

**S4 Fig. Separation of classes in PC1 and 2.** Estimated using indels (**a**) and substitutions (**b**).
(TIF)

**S5 Fig. Subclasses of OC43 were separated by the indicated PCs found in the whole genome.** Conservative characteristics of the virus and repeated appearance were obvious.
(TIF)

**S6 Fig. a.** Annual changes in the MERS-CoV genome**. b.** Comprehensive data for SARS-CoV-2. Samples found in some European countries showed higher magnitudes of PCs, indicating accumulation of mutations (also see S4 Table).
(TIF)

**S1 Table. PC for samples of whole genome sequences of coronaviruses.** Fig 1 was made from a part of this table.
(XLSX)

**S2 Table Differences in spike proteins a SARS-CoV and bat SARS-like viruses b SCoV2 and other coronaviruses Aligned protein sequences were compared The IDs for the samples are listed.**
(XLSX)

**S3 Table. PC for samples of HCoV OC43 coronaviruses.**
(XLSX)

**S4 Table. PCs of SARS-CoV-2.**
(XLSX)

**S1 Data. Aligned sequences for the coronaviruses.**
(ZIP)

**S2 Data. Aligned sequences for the OC43.**
(ZIP)

**S3 Data. Aligned sequences for SARS-CoV-2.**
(ZIP)

## Acknowledgments

The author would like to thank Editage (www.editage.com) for English language editing.

## Author Contributions

**Conceptualization:** Tomokazu Konishi.

**Data curation:** Tomokazu Konishi.

**Formal analysis:** Tomokazu Konishi.

**Funding acquisition:** Tomokazu Konishi.

**Investigation:** Tomokazu Konishi.

**Methodology:** Tomokazu Konishi.

**Project administration:** Tomokazu Konishi.

**Resources:** Tomokazu Konishi.

**Software:** Tomokazu Konishi.

**Supervision:** Tomokazu Konishi.

**Validation:** Tomokazu Konishi.

**Visualization:** Tomokazu Konishi.

**Writing – original draft:** Tomokazu Konishi.

**Writing – review & editing:** Tomokazu Konishi.

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
