## [Decision Letter · Decision Letter 0]

6 Aug 2020

PONE-D-20-16191

Coronavirus, as a source of pandemic pathogens.

PLOS ONE

Dear Dr. Konishi,

Thank you for submitting your manuscript to PLOS ONE. After careful consideration, we feel that it has merit but does not fully meet PLOS ONE’s publication criteria as it currently stands. Therefore, we invite you to submit a revised version of the manuscript that addresses the points raised during the review process.

I believe all suggested modifications can be done by some minor corrections or addressing them in the manuscript. You are requested to make these changes and resubmit your manuscript so we can reach to a decision on your article soon.

We look forward to receiving your revised manuscript.

Kind regards,

Binod Kumar, PhD

Academic Editor

PLOS ONE

Journal Requirements:

2.We suggest you thoroughly copyedit your manuscript for language usage, spelling, and grammar. If you do not know anyone who can help you do this, you may wish to consider employing a professional scientific editing service.  

Reviewers' comments:

Reviewer's Responses to Questions

**Comments to the Author**

1. Is the manuscript technically sound, and do the data support the conclusions?

Reviewer #1: Partly

Reviewer #2: Yes

2. Has the statistical analysis been performed appropriately and rigorously? 

Reviewer #1: Yes

Reviewer #2: No

3. Have the authors made all data underlying the findings in their manuscript fully available?

Reviewer #1: Yes

Reviewer #2: Yes

4. Is the manuscript presented in an intelligible fashion and written in standard English?

Reviewer #1: No

Reviewer #2: No

5. Review Comments to the Author

Reviewer #1: COMMENTS TO AUTHORS

The manuscript entitled “Coronavirus, as a source of pandemic pathogens” deals with comparison of the genomes of influenza and human coronaviruses in an attempt to understand the current transmission of the pandemic SARS-CoV-2 and also the future patterns of circulation. The findings seem to be appropriate in the present context. The author comprehensively links the circulation patterns with the type of genomes through the PCA done in this study. Important observations have also been recorded that may be considered for better preparedness for any potential threat in future. However, the manuscript requires extensive revision for better clarity of the research work.

I have the following suggestions on the manuscript.

1. Lines 28-29: Please add a reference to this sentence. The meaning is not being understood. How the author concludes that the limited variations in CoVs “eased the novel transfection of CoV to humans” needs to be elaborated a little.

2. Line 37: it must be “nearly 30 kb”.

3. Lines 37-38: what about influenza genome. It is single stranded RNA but negative sense segmented genome of about 3.5 kb size. The comparative parameters must be same in one sentence for an apt comparison.

4. Lines 39-40: Add a reference to this statement.

5. Lines 43-46: Reference for an “another large study” is also same, i.e. ref. no. 6. Pl. check the appropriate ref.

Also, here it is important to mention the current infectivity and case fatality rates for human CoVs and influenza viruses.

6. Lines 55- 56, 109 and others: Full forms for abbreviations are required at places of first mention: DDBJ, DECIPHER, TGEV, etc. throughout the manuscript.

7. Materials and Methods: This section needs extensive revision. What were the study samples or groups and their sources have not been indicated? Were sequences of all the three, i.e., SARS-CoV, MERS-CoV and SARS-CoV-2 were analyzed?

8. Line 57: What kind of “samples”?

9. Line 64: Were only 2796 full-length data available or this much were narrowed down based on any particular parameters? If the latter, explain those parameters used for short-listing the desired sequences. And why others were excluded.

10. The methodology section should be made very clear to make it understandable for even a novice in the area. It is especially more important for such an article that addresses the current COVID-19 pandemic.

11. It has been stated that the CoVs mutate less but evolve more (add ref.). Please discuss about this also.

12. Line 128: Please revise this sentence for aptness. Human outbreak strains must be replaced with more appropriate scientific terminology, for eg., Human CoVs of epidemic potential or epidemic human CoVs.

13. Line 129: “SARS spike protein” to be replaced with “spike protein of SARS-CoV”.

14. The scientific aptness for usage of words needs to be carefully proof-read throughout the manuscript.

15. Lines 135- 136: Can this statement be supported by any other epidemiological study (studies based on respiratory clinical specimens and sero-surveillance) reference?

16. Lines 138-139: Kindly re-frame the sentence for better clarity as this seems one of the important observations of the study. For eg., what “variety” is being referred to here?

17. Even though the data is depicted in figures, the text should maintain clarity of thoughts of the authors.

18. Line 147: contrastive to be replaced by “in contrast”.

19. The manuscript needs to be extensively revised for usage of English language in order to aptly express the important scientific findings of this work.

20. Line 152: “Emvecovirus” to be corrected “Embecovirus”.

21. Lines 159-162: Important statement, but requires revision for clarity. For instance, it can be “The spike protein mediated infection by CoVs…more adaptable…”

22. Line 176-177: “Adaptation to the genetic system of a new host may alter codon usage and several amino acids” needs reference.

23. Line 185: “SCoV2 satisfies all these conditions…” needs reference.

24. Line 188: “variable residues during three decades”. Which residues are being talked about and the time span? If possible, elaborate a bit more.

25. Line 190: “infectibility” or infectivity?

26. Line 191: “influenza A did.” To be replaced with “influenza A virus did.”

27. Line 198: “The ORF lengths for the influenza virus are within a certain range...” What is this range? Add a reference also. This comparison of ORFs must be further expanded for better understanding of the readers.

28. Line 203: “Spike protein… and may cause antibody-dependent enhancement”. Why and how? Must be elaborated here with proper references.

29. The comparison of coronavirus and influenza viruses still requires to be addressed in a more extensive manner in relation to the present findings. For instance, lines 198- 203: can an example of influenza virus protein be taken here for comparison?

30. Lines 224- 228: How can this issue be addressed. Elaborate a bit more and if possible include this in a separate section for conclusion.

Reviewer #2: 1. Title of manuscript is not very apt for the content of manuscript. I would suggest to select more apt title.

2. Line 33- “Rather, those repeatedly found among humans showed annual changes”. Cite a reference for this sentence.

3. Line 44- “ a 2010-2015 study in China reported that 2.3% and 30% of patients were positive for coronavirus and

influenza virus, respectively [6]; a similar ratio was found in another large study 46 [6].

Please provide correct referencing for larger study mentioned in the sentence.

4. Full forms of abbreviations used are missing.

5. Materials and methods need extensive revision to indicate type of sample used, which all sequences were compared

and analyzed. Software and tools used to compare and conclude the findings.

6. Line 104-110 sentences need to be reframed and structured for clarity of readers. It is not clear as a sentence talks

about influenza and suddenly next sentence seems to be talking about coronavirus.

7. Line 114 “Similarly to other RNA viruses [5], many indels were observed, especially in some smaller ORFs” replace

similarly with similar.

8. Many long sentences have been used. For the clarity of readers reframe sentences.

9. Line 128 “Each of the human-outbreak strains had similar ones in bats or camels, with minor differences 129 (Fig. 1

and S1 Table)”. Please correct it grammatically

10. The word contrarily has been used many a times. Use other forms of the word.

11. Line 158- These characteristics corroborate the assessment that “coronaviruses can apparently breach cell type,

tissue, and host species barriers with relative ease. Relative to what?

12. Authors must clearly mention why they are comparing influenza and Coronavirus. How that may help in curbing

present pandemic.

13. Figures should be discussed extensively in the result and discussion section.

14. Results need extensive discussion as at many places information is missing or is not very clear.

15. Thorough revision of english language and sentence reformation is required.

16. Reference number 15 and 17 need formatting

6. PLOS authors have the option to publish the peer review history of their article (what does this mean?). If published, this will include your full peer review and any attached files.

Reviewer #1: No

Reviewer #2: No

---

## [Author Response · Author response to Decision Letter 0]

28 Aug 2020

>1. Lines 28-29: Please add a reference to this sentence. The meaning is not being understood. How the author concludes that the limited variations in CoVs “eased the novel transfection of CoV to humans” needs to be elaborated a little.

I deleted Lines 28-29 to satisfy a request by reviewer 2. It was a summary of the article. Instead, the conclusions were drawn at the end of the Discussion.

Additionally, I altered the sentence to explain why the new strain will infect humans with ease. I hope that the meaning is now clear.

>2. Line 37: it must be “nearly 30 kb”.

I changed the sentence accordingly.

>3. Lines 37-38: what about influenza genome. It is single stranded RNA but negative sense segmented genome of about 3.5 kb size. The comparative parameters must be same in one sentence for an apt comparison.

I altered the paragraph and explained the basis of comparisons, adding to the explanation of comparison in the materials and methods.

>4. Lines 39-40: Add a reference to this statement.

I added a review article that explains that RNA-dependent RNA polymerase has lower fidelity and how this contributes to the evolution of the virus.

>5. Lines 43-46: Reference for an “another large study” is also same, i.e. ref. no. 6. Pl. check the appropriate ref.

I apologize for the wrong citation. I have corrected it.

Also, here it is important to mention the current infectivity and case fatality rates for human CoVs and influenza viruses.

Fatality rates are still difficult to compare. However, I mentioned this and cited new references.

>6. Lines 55- 56, 109 and others: Full forms for abbreviations are required at places of first mention: DDBJ, DECIPHER, TGEV, etc. throughout the manuscript.

I have added the full forms for DDBJ, TGEV, SARS, MARS, HCoV, and COVID. However, DECIPHER is not an abbreviation but a unique noun given by the developer “DECIPHER is a software toolset that can be used for deciphering and managing biological sequences efficiently using the R programming language.” 

>7. Materials and Methods: This section needs extensive revision. What were the study samples or groups and their sources have not been indicated? Were sequences of all the three, i.e., SARS-CoV, MERS-CoV and SARS-CoV-2 were analyzed?

All the sequences used are available in the Supporting Information files. As GISAID did not allow us to share the sequences, only the list was given.

>8. Line 57: What kind of “samples”?

I added the term “virus.”

>9. Line 64: Were only 2796 full-length data available or this much were narrowed down based on any particular parameters? If the latter, explain those parameters used for short-listing the desired sequences. And why others were excluded.

As was written, all 2796 data points were used in Fig 3C. One of the advantages of PCA is that it is capable of handling a large number of samples. Now, I am submitting a new article, which uses many more virus samples to see the present state of the pandemic.

>10. The methodology section should be made very clear to make it understandable for even a novice in the area. It is especially more important for such an article that addresses the current COVID-19 pandemic.

I added a paragraph explaining PCA. I hope that this gives enough information. 

>11. It has been stated that the CoVs mutate less but evolve more (add ref.). Please discuss about this also.

I am afraid there was a misinterpretation. HCoVs are conservative (Fig. 3A) and, with the current data, the evolutionary rate of SARS-CoV-2 cannot be estimated. Coronaviruses have not evolved rapidly when compared with influenza (Fig. 1, S2A Fig.). 

HCoVs are more conservative than flu viruses (Fig. 3A). The annual change is very small (Fig. 3B) in comparison with the influenza N1H1. To clarify this, I added a Supporting figure (S2G Fig.) where the difference in the evolutionary rate is very clear.

>12. Line 128: Please revise this sentence for aptness. Human outbreak strains must be replaced with more appropriate scientific terminology, for eg., Human CoVs of epidemic potential or epidemic human CoVs.

I replaced the term accordingly.

>13. Line 129: “SARS spike protein” to be replaced with “spike protein of SARS-CoV”.

I replaced the wording accordingly. 

>14. The scientific aptness for usage of words needs to be carefully proof-read throughout the manuscript.

Unfortunately, English is not my first language, and so as was stated, the manuscript was professionally edited for language before submission. I have reordered English editing. I hope that this improves the manuscript.

>15. Lines 135- 136: Can this statement be supported by any other epidemiological study (studies based on respiratory clinical specimens and sero-surveillance) reference?

Unfortunately, no. The differences in the viruses are not detectable by ordinary clinical or serum analysis. Only DNA sequencing can reveal differences. However, for a long time, we used the wrong methods for sequence analysis; this might have obscured many obvious pieces of evidence. I refer to a new figure added in the Supporting Information.

>16. Lines 138-139: Kindly re-frame the sentence for better clarity as this seems one of the important observations of the study. For eg., what “variety” is being referred to here?

I have added explanations to clarify what was selected.

I wish to thank the reviewer for the comment. I found that the European sample (Normandy, France) was reported in 2017, but it was collected in 2002. I corrected Fig 2A and 2B accordingly.

>17. Even though the data is depicted in figures, the text should maintain clarity of thoughts of the authors.

I have reordered the English edition. I hope that this improves the manuscript.

>18. Line 147: contrastive to be replaced by “in contrast”.

I replaced the wording accordingly. 

>19. The manuscript needs to be extensively revised for usage of English language in order to aptly express the important scientific findings of this work.

I have reordered English editing. I hope that this improves the manuscript.

>20. Line 152: “Emvecovirus” to be corrected “Embecovirus”.

I replaced the wording accordingly. 

>21. Lines 159-162: Important statement, but requires revision for clarity. For instance, it can be “The spike protein mediated infection by CoVs…more adaptable…”

I replaced the wording accordingly. 

>22. Line 176-177: “Adaptation to the genetic system of a new host may alter codon usage and several amino acids” needs reference.

I added a reference.

>23. Line 185: “SCoV2 satisfies all these conditions…” needs reference.

I added a reference.

>24. Line 188: “variable residues during three decades” Which residues are being talked about and the time span? If possible, elaborate a bit more.

I added an explanation for the residues and why those residues seem to have changed.

>25. Line 190: “infectibility” or infectivity?

I wish to thank the reviewer. It was infectibility. 

>26. Line 191: “influenza A did.” To be replaced with “influenza A virus did.”

I replaced the wording accordingly. 

>27. Line 198: “The ORF lengths for the influenza virus are within a certain range...” What is this range? Add a reference also. This comparison of ORFs must be further expanded for better understanding of the readers.

I have added a reference and explanations.

28. Line 203: “Spike protein… and may cause antibody-dependent enhancement”. Why and how? Must be elaborated here with proper references.

I wish to thank the reviewer for the comment. Here, I cited a wrong reference. I corrected the reference and added a new one related to the phenomenon “antibody dependent enhancement.”

29. The comparison of coronavirus and influenza viruses still requires to be addressed in a more extensive manner in relation to the present findings. For instance, lines 198- 203: can an example of influenza virus protein be taken here for comparison?

I wish to thank the reviewer for this comment. Instead of taking a protein as an example, I explained how all proteins changed. In addition, I added information on the length of the ORFs to make the explanations tangible.

30. Lines 224- 228: How can this issue be addressed. Elaborate a bit more and if possible include this in a separate section for conclusion.

I added an explanation for the methodology and explained the real classification of coronaviruses.

Reviewer #2: 1. Title of manuscript is not very apt for the content of manuscript. I would suggest to select more apt title.

I added a term “seasonal” The current title should be suitable for the summary and the concluding paragraph newly placed at the end of the discussion section. 

2. Line 33- “Rather, those repeatedly found among humans showed annual changes”. Cite a reference for this sentence.

I have cited a reference here. 

3. Line 44- “ a 2010-2015 study in China reported that 2.3% and 30% of patients were positive for coronavirus and

influenza virus, respectively [6]; a similar ratio was found in another large study 46 [6].

Please provide correct referencing for larger study mentioned in the sentence.

I wish to thank the reviewer for this comment. I apologise for the incorrect citation. I corrected it. 

4. Full forms of abbreviations used are missing.

I have corrected this accordingly. 

5. Materials and methods need extensive revision to indicate type of sample used, which all sequences were compared

and analyzed. Software and tools used to compare and conclude the findings.

All the samples were examined using the same method, as has been written. I added an explanation at the beginning of the paragraph.

6. Line 104-110 sentences need to be reframed and structured for clarity of readers. It is not clear as a sentence talks

about influenza and suddenly next sentence seems to be talking about coronavirus.

Throughout the Results section, what was discussed was coronavirus. Examples of influenza were used for comparisons with coronavirus. To clarify this, I inserted the term “coronavirus” in every sentence that contains the names of the subclasses in the Results section.

7. Line 114 “Similarly to other RNA viruses [5], many indels were observed, especially in some smaller ORFs” replace similarly with similar.

I replaced the wording accordingly.

8. Many long sentences have been used. For the clarity of readers reframe sentences.

I checked the length of all sentences. 

Some were compound sentences. Although they are not harder to understand, I changed those into single sentences (including sentences separated by semicolons). 

Some had a non-restrictive relative clause. Those are difficult to separate and should not cause reading difficulty.

The average number of words per sentence was 15.37. This is within the range (12-17) that is asked for scientific writing https://www.aje.com/arc/editing-tip-sentence-length/. 

I have re-ordered English editing; I hope this improves the readability.

9. Line 128 “Each of the human-outbreak strains had similar ones in bats or camels, with minor differences 129 (Fig. 1 and S1 Table)”. Please correct it grammatically

I altered the wording.

10. The word contrarily has been used many a times. Use other forms of the word.

The manuscript used the word “contrarily” three times. I have used “in contrast” four times, so maybe this should be avoided. I replaced two instances with “on the other hand” and “to the contrary.”

11. Line 158- These characteristics corroborate the assessment that “coronaviruses can apparently breach cell type, tissue, and host species barriers with relative ease. Relative to what?

The authors of this article did not specify it. However, they compared coronaviruses with influenza or ebola viruses. So, I added “other viruses.” 

12. Authors must clearly mention why they are comparing influenza and Coronavirus. How that may help in curbing present pandemic.

I added some information about influenza and quoted a phrase of Sun Tzu. 

13. Figures should be discussed extensively in the result and discussion section.

I deleted a part of the Introduction section, which briefly introduced the manuscript; this deleted explanations about the Figures. Some text in the Materials and Methods just show cues to understanding, so they do not use for discussions. I believe that this helps with readability.

14. Results need extensive discussion as at many places information is missing or is not very clear.

I have added some Supporting Figures. I hope that this will improve clarity.

15. Thorough revision of english language and sentence reformation is required.

I requested a professional re-editing of the manuscript. I hope that this will improve the English. 

16. Reference number 15 and 17 need formatting

I deleted some letters “_” from both of them.

---

## [Decision Letter · Decision Letter 1]

16 Oct 2020

PONE-D-20-16191R1

Principal component analysis of coronaviruses reveals their diversity and seasonal and pandemic potential.

PLOS ONE

Dear Dr. Konishi,

Thank you for submitting your manuscript to PLOS ONE. After careful consideration, we feel that it has merit but does not fully meet PLOS ONE’s publication criteria as it currently stands. Therefore, we invite you to submit a revised version of the manuscript that addresses the points raised during the review process.

We look forward to receiving your revised manuscript.

Kind regards,

Binod Kumar, PhD

Academic Editor

PLOS ONE

Additional Editor Comments (if provided): Please address the minor comment from reviewer 1 by text additions in the manuscript. No major changes are allowed to be made at this stage.

Reviewers' comments:

Reviewer's Responses to Questions

**Comments to the Author**

1. If the authors have adequately addressed your comments raised in a previous round of review and you feel that this manuscript is now acceptable for publication, you may indicate that here to bypass the “Comments to the Author” section, enter your conflict of interest statement in the “Confidential to Editor” section, and submit your "Accept" recommendation.

Reviewer #1: (No Response)

Reviewer #2: All comments have been addressed

2. Is the manuscript technically sound, and do the data support the conclusions?

Reviewer #1: Yes

Reviewer #2: Yes

3. Has the statistical analysis been performed appropriately and rigorously? 

Reviewer #1: N/A

Reviewer #2: Yes

4. Have the authors made all data underlying the findings in their manuscript fully available?

Reviewer #1: Yes

Reviewer #2: Yes

5. Is the manuscript presented in an intelligible fashion and written in standard English?

Reviewer #1: Yes

Reviewer #2: Yes

6. Review Comments to the Author

Reviewer #1: In response to Previous Comment #9 by Reviewer 1:

Line 64: Were only 2796 full-length data available or this much were narrowed

down based on any particular parameters? If the latter, explain those parameters used

for short-listing the desired sequences. And why others were excluded.

Author's response: As was written, all 2796 data points were used in Fig 3C. One of the advantages of

PCA is that it is capable of handling a large number of samples. Now, I am submitting a

new article, which uses many more virus samples to see the present state of the

pandemic.

The response is not satisfactory. The question was that whether only 2796 sequences were available for the analysis, or this number was shortlisted using any specific criteria for the present work?

Kindly respond for the same.

The manuscript has been extensively revised for better understanding, both scientifically and for use of English language. However, the manuscript still has scope of improvement in terms of usage of English language in different portions of the revised manuscript.

Reviewer #2: Authors have appropriately answered the queries and improved the manuscript. I thus recommend the publication of the manuscript in the present format.

7. PLOS authors have the option to publish the peer review history of their article (what does this mean?). If published, this will include your full peer review and any attached files.

Reviewer #1: **Yes: **Roopali Rajput

Reviewer #2: No

---

## [Author Response · Author response to Decision Letter 1]

28 Oct 2020

>The response is not satisfactory. The question was that whether only 2796 sequences were available for the analysis, or this number was shortlisted using any specific criteria for the present work?

>Kindly respond for the same.

It was unfortunate that the intended meaning was not clear. As I wrote in the original manuscript and the response, it was ALL the sequences available at that time.

>The manuscript has been extensively revised for better understanding, both scientifically and for use of English language. However, the manuscript still has scope of improvement in terms of usage of English language in different portions of the revised manuscript.

I really appreciate the reviewer pointed the problems; those were so helpful. I revised the manuscript accordingly. One-to-one responses were added to the revised form. I hope those improve the readability.

---

## [Decision Letter · Decision Letter 2]

13 Nov 2020

Principal component analysis of coronaviruses reveals their diversity and seasonal and pandemic potential.

PONE-D-20-16191R2

Dear Dr. Konishi,

We’re pleased to inform you that your manuscript has been judged scientifically suitable for publication and will be formally accepted for publication once it meets all outstanding technical requirements.

Kind regards,

Binod Kumar, PhD

Academic Editor

PLOS ONE

Additional Editor Comments (optional):

Reviewers' comments:

Reviewer's Responses to Questions

**Comments to the Author**

1. If the authors have adequately addressed your comments raised in a previous round of review and you feel that this manuscript is now acceptable for publication, you may indicate that here to bypass the “Comments to the Author” section, enter your conflict of interest statement in the “Confidential to Editor” section, and submit your "Accept" recommendation.

Reviewer #1: All comments have been addressed

2. Is the manuscript technically sound, and do the data support the conclusions?

Reviewer #1: Yes

3. Has the statistical analysis been performed appropriately and rigorously? 

Reviewer #1: Yes

4. Have the authors made all data underlying the findings in their manuscript fully available?

Reviewer #1: Yes

5. Is the manuscript presented in an intelligible fashion and written in standard English?

Reviewer #1: Yes

6. Review Comments to the Author

Reviewer #1: (No Response)

7. PLOS authors have the option to publish the peer review history of their article (what does this mean?). If published, this will include your full peer review and any attached files.

Reviewer #1: No

---

## [Editor Report · Acceptance letter]

18 Nov 2020

PONE-D-20-16191R2 

Principal component analysis of coronaviruses reveals their diversity and seasonal and pandemic potential. 

Dear Dr. Konishi:

I'm pleased to inform you that your manuscript has been deemed suitable for publication in PLOS ONE. Congratulations! Your manuscript is now with our production department. 

Kind regards, 

on behalf of

Dr. Binod Kumar 

Academic Editor

PLOS ONE